# Rett-like Phenotypes in *HNRNPH2*-Related Neurodevelopmental Disorder

**DOI:** 10.3390/genes14061154

**Published:** 2023-05-26

**Authors:** Joseph Nicho Gonzalez, Sylvie Goldman, Melissa T. Carter, Jennifer M. Bain

**Affiliations:** 1Vagelos College of Physicians and Surgeons, Columbia University, New York, NY 10032, USA; jng2147@cumc.columbia.edu; 2Department of Neurology, Division of Child Neurology, Columbia University Irving Medical Center, New York, NY 10032, USA; 3Sergievsky Center, Columbia University Irving Medical Center, New York, NY 10032, USA; 4Department of Genetics, Children’s Hospital of Eastern Ontario, Ottawa, ON K1H 8L1, Canada; 5Morgan Stanley Children’s Hospital, NewYork Presbyterian Hospital, New York, NY 10032, USA

**Keywords:** *HNRNPH2*, *HNRNPH2*-related neurodevelopmental disorder (*HNRNPH2*-RNDD), neurodevelopmental disorder (NDD), Rett Syndrome, neurodevelopmental disorder, Rett-like phenotypes

## Abstract

Rett Syndrome (RTT) is a neurodevelopmental disorder with a prevalence of 1:10,000 to 15,000 females worldwide. Classic Rett Syndrome presents in early childhood with a period of developmental regression, loss of purposeful hand skills along with hand stereotypies, gait abnormalities, and loss of acquired speech. Atypical RTT is diagnosed when a child shows some but not all the phenotypes of classic RTT, along with additional supporting criteria. Over 95% of classic RTT cases are attributed to pathogenic variants in Methyl-CpG Binding Protein 2 (*MECP2*), though additional genes have been implicated in other RTT cases, particularly those with the atypical RTT clinical picture. Other genetic etiologies have emerged with similar clinical characteristics to RTT Syndrome. Our team has characterized *HNRNPH2*-related neurodevelopmental disorder (*HNRNPH2-*RNDD) in 33 individuals associated with de novo pathogenic missense variants in the X-linked *HNRNPH2* gene, characterized by developmental delay, intellectual disability, seizures, autistic-like features, and motor abnormalities. We sought to further characterize RTT clinical features in this group of individuals by using caregiver report. Twenty-six caregivers completed electronic surveys, with only 3 individuals having previously received an atypical RTT diagnosis, and no individuals with a typical RTT diagnosis. Caregivers reported a high number of behaviors and/or phenotypes consistent with RTT, including the major criteria of the syndrome, such as regression of developmental skills and abnormal gait. Based on the survey results, 12 individuals could meet the diagnostic clinical criteria for atypical RTT Syndrome. In summary, individuals with *HNRNPH2-*RNDD exhibit clinical characteristics that overlap with those of RTT, and therefore, *HNRNPH2-*RNDD, should be considered on the differential diagnosis list with this clinical picture.

## 1. Introduction

In 1966, Andreas Rett, an Austrian pediatric neurologist, first described a neurodevelopmental disorder presenting in childhood with a period of regression followed by recovery or stabilization, loss of purposeful hand skills along with the presentation of hand stereotypies (i.e., hand wringing), gait abnormalities, and loss of spoken language, subsequently referred to as Rett Syndrome (RTT) [1]. Hagberg et al. further characterized the clinical description in the early 80s [2], followed by identification of the causative gene, *MECP2*, by Amir et al. in 1999 [3]. Neul expanded the definition of Rett Syndrome to include both “typical” Rett Syndrome and “atypical” Rett Syndrome [4] (Figure 1). Currently, the diagnosis of RTT requires meeting a set of clinical diagnostic criteria described by Neul et al. in 2010, regardless of the presence of a *MECP2* pathogenic variant. For a diagnosis of classic RTT, an individual must fulfill four major clinical criteria, along with evidence of regression followed by a period of stabilization while atypical RTT requires the presence of at least 2 of the 4 major criteria, and at least 5 of the 11 supportive criteria. 

Rett Syndrome is a progressive disorder with no known cure and is characterized by four well-defined stages [5]. The first stage typically presents between 6–18 months, with slight behavioral abnormalities characterizing this stage, which include loss of interest in toys the child previously liked, repetitive hand movements that do not disrupt normal functioning, and often with less eye contact and engagement with caregivers [5]. It is not uncommon for a clinician to suspect these behavioral abnormalities as a presentation of autism spectrum disorder (ASD), as a notable percentage of ASD cases present with early regression of skills [4]. Stage 2 typically presents between 1 and 4 years of age with more rapid loss of hand skills and spoken language. Behavioral abnormalities are more prominent in this stage, and the classic “hand-wringing” stereotypies emerge. Stage 3 presents around late childhood to adolescence with symptoms such as ataxia, apraxia, seizures, handwringing, and loss of spoken language, which often stabilize. Stage 4 presents in adulthood with further loss of motor skills, in addition to possible new-onset scoliosis, increased muscle weakness, spasticity, rigidity, and abnormal posturing [6]. Not all individuals progress to Stage 4, but all individuals with RTT can have a debilitating disorder presenting with many challenges and without any targeted treatments. 

While initial studies showed that pathogenic variants in the *MECP2* gene accounted for around 95–97% of individuals with typical Rett Syndrome [7], the remainder of subjects met the clinical criteria for typical Rett Syndrome without any identifiable genetic variant within *MECP2* [7]. Absence of *MECP2* mutations in RTT patients is even more apparent when considering only atypical RTT diagnoses, as only 50–70% of these cases have identified mutations in *MECP2* [8]. The clinical presentation of *MECP2* mutations is broad, as some young females with pathogenic *MECP2* genetic changes present with only minor cognitive problems, such as a learning disability or difficult-to-control aggression, and do not meet the full clinical criteria for typical or atypical RTT [9,10]. There have even been cases of pathologic *MECP2* variants in children diagnosed with autism spectrum disorder without a comorbid RTT diagnosis [11]. Finally, there is some evidence that suggests that overexpression of *MECP2* gene may be detrimental to the developing nervous system, as mice with *MECP2* overexpression were found to have seizures, behavioral problems, and shortened lifespans, implying that there may be a gene dosage effect on clinical phenotypes [12].

With the field of clinical and research genetic testing rapidly expanding, other genetic variants have been identified in individuals with RTT phenotypes, such as genes affecting brain development like *CDKL5* and *FOXG1* [5,8]. *CDKL5* modulates neuronal growth, differentiation, and migration, while *FOXG1* works as a transcription repressor in genetic pathways involved in neuronal development. More than 80 pathogenic *CDKL5* variants have been reported in females with Rett-like features [13,14]. Another prominent feature in these subjects is early-onset epileptic encephalopathy, with seizures presenting before 5 months of age, along with severe cognitive impairment and absent speech. Head growth deceleration is also prominent in these subjects [13]. More recently, 7 males with severe epileptic encephalopathy have been described with *CDKL5* deficiency disorder [15]. Far fewer cases of *FOXG1-*related disorder have been described in the literature, with the existent population estimated at less than 20 [15]. *FOXG1* pathogenic variants can lead to dysmorphic features, tone abnormalities, and autonomic features, such as small, cold hands and feet and peripheral vasomotor disturbances [15].

Our group first identified the X-linked RNA-binding protein, *HNRNPH2* (heterogeneous nuclear ribonucleoprotein H2) to be causative of a neurodevelopmental disorder, with some individuals reporting a formal RTT diagnosis from a clinician provider [16,17]. *HNRNPH2* encodes the HNRNPH2 protein which is part of a larger family of a ubiquitously expressed RNA-binding proteins called heterogeneous nuclear ribonucleoproteins (hnRNPs) [18]. HnRNPs have multiple roles in pre-mRNA processing, affecting mRNA metabolism and transportation within cells. They are considered master orchestrators of development, integral to producing and expressing different protein products in the brain and throughout the body [18]. We initially described the clinical phenotype and have expanded on the natural history of *HNRNPH2*-RNDD (OMIM 300986, Intellectual developmental disorder, X-linked syndromic, Bain-type), presenting with developmental delay/intellectual disability, severe language impairment, motor problems, growth, and musculoskeletal disturbances [16,17,19]. Less common findings include dysmorphic features, epilepsy, cortical visual impairment and neuropsychiatric diagnoses such as autism spectrum disorder and anxiety [16,17,19]. After Peron et al. published a case report about an individual with a Rett diagnosis and *HNRNPH2*-RNDD, we noted several other individuals in our cohort who had reported Rett-like phenotypes as well [20]. In addition, during in-person assessments, the clinical team noted stereotypic hand movements in several individuals. As such, we were interested in better describing whether there are other RTT-like clinical features in individuals with *HNRNPH2*-RNDD.

## 2. Materials and Methods

Individuals were recruited from the *HNRNPH2* Natural History Study (NHS) (NCT03492060). Inclusion criteria for this NHS require a genetic confirmation of a pathogenic or likely pathogenic variant in *HNRNPH2* using American College of Medical Genetics and Genomics/Association of Molecular Pathology criteria, largely identified via clinical exome sequencing. This NHS study is approved by the Columbia University Institutional Review Board, and informed consent was obtained from all caregivers or legal guardians. Individuals were sent electronic surveys through Research Electronic Data Capture (REDCap), which consisted of 38 questions for the initial survey and a follow-up survey of 20 questions. The survey was only available in English. We collected basic demographic information such as age, sex, genotype, parental ancestry, presence (or absence) of RTT clinical diagnosis, and current growth parameters. Next, we asked caregivers to indicate the presence or absence of signs and symptoms consistent with the RTT Syndrome diagnostic criteria (Neul’s criteria, both main and supportive criteria, summarized in Figure 1). We collected data about other medical diagnoses such as seizures. A follow up survey was sent to all individuals to clarify regressions that had been previously reported. Finally, we calculated the proportion of individuals who met Neul’s major and minor diagnostic clinical criteria and reported them below.

## 3. Results

Thirty individuals completed the initial survey from the entire registry with 50 individuals. After exclusion for incomplete survey responses, we had 26 total surveys for final analyses, including 23 females, and 3 males (Table 1). Notably, data from two individuals who have passed away, at 12 and 36 years old, were included in the analyses. The age range of affected individuals was 3 to 42 with the average and median age of the probands were 13.6 years and 8 years, respectively. The cohort comprised a variety of genetic variants including: Arg206Trp (n = 16), p.Arg206Gln (n = 3), p.Arg114Trp (n = 2), p.Arg188Ter (n = 2), p.Pro209Leu (n = 1), p.Arg212Ser (n = 1), and p.Arg212Thr (n = 1). A majority of the participants were from Caucasian/white ethnicity (n = 23), with one Asian and 2 Hispanic participants (Table 1).

We recorded the number of major and minor criteria for each subject. (Table 2 and Table 3). We grouped individuals into subgroups based on the amount of major and minor criteria (Table 3). Twelve individuals met clinical criteria for atypical RTT (i.e., 2 or more major criteria, 5 or more supportive criteria), though only three individuals (3/26 or 11.5 %) had been previously diagnosed with atypical RTT by their clinical providers.

All but one subject presented with developmental delays. Seventeen caregivers (17/26, 65%) reported a period of regression followed by recovery or stabilization. Of note, the 3 participants clinically diagnosed with Atypical RTT did not report any regression.

Fourteen caregivers 14/26 (54%) reported the presence of comorbid seizures, and many of these individuals reported multiple Rett-like characteristics. Six caregivers reported self-harm (6/26 or 23%). Of these individuals, one met 3 main criteria and 3 supportive criteria, and three additional individuals met 2 major criteria with 5 or more supportive criteria.

## 4. Discussion

*HNRNPH2*-RNDD is a newly characterized disorder caused by pathogenic variants in the X-linked gene *HNRNPH2* which encodes for a ubiquitously expressed RNA-binding protein HNRNPH2 [16,17,19,21,22]. Affected individuals usually have a history of global developmental delay and intellectual disability in addition to significant motor disturbances and language impairment. Many individuals also have epilepsy, cortical visual impairment, and neuropsychiatric diagnoses such as autism spectrum disorder and anxiety. We have previously shown significantly delayed motor skills in individuals with an *HNRNPH2*-RNDD, but we also noted the presence of motor stereotypies in these individuals [16]. Peron et al. reported a case study of an adult individual with a clinical diagnosis of atypical RTT and some caregivers in the NHS reported having this diagnosis [20]. In order to better elucidate the presence Rett-like characteristics in *HNRNPH2*-related NDD, we surveyed caregivers about the clinical characteristics of individuals using the clinical criteria most commonly used for RTT diagnoses (typical and atypical) by Neul et al. [4]. At the time of this survey, our preliminary registry included fifty probands and we had 26 completed surveys.

Three of the twenty-six individuals had been previously given a clinical diagnosis of RTT Syndrome, however, our survey did not support this. All three of these individual’s parents did not report regression, which is required for typical or atypical Rett diagnosis. However, there certainly are similarities between phenotypes presenting in *HNRNPH2-*RDD and in RTT. In the overall group of respondents, the majority (18/26 or 69%) of the individuals met at least 2 of Neul’s major criteria. The most common major criteria were gait abnormalities and hand stereotypies, which were both noted by the clinical research team during NHS evaluations. The most common supportive criteria were tone abnormalities and inappropriate laughter/screaming spells.

For the minor criteria, we describe some notable trends below. Twenty-five individuals (25/26) reported tone abnormalities, which was also noted during the neurological examinations for the NHS. It remains unclear if these tone abnormalities, mostly hypotonia but some hypertonia, are causal or correlated to the gait abnormalities, as 21 of the 25 individuals with abnormal tone report also having gait abnormalities. Only 24 individuals provided responses about pain, with thirteen reporting diminished response to pain. Thirteen individuals had small, cold hands/feet. We found 12/13 individuals with small, cold hands/feet had abnormal gait, and 12/13 individuals with diminished response to pain also had abnormal gait. In addition, 12/13 individuals with abnormal gait, small, cold hands/feet and/or diminished response to pain had abnormal muscle tone as well. Again, it remains unclear whether there is a relationship between pain, vasomotor symptoms of extremities, abnormal muscle tone and abnormal gait.

Some other results deserve special mention. Self-harm was reported in 6 individuals, three of whom met criteria for atypical Rett Syndrome. The mechanism underlying self-harm behaviors is beyond the scope of this paper, and the overall incidence and prevalence of self-harm behaviors in RTT has not been well categorized, yet others have suggested self-harm behaviors may be related to sensory processing abnormalities [23]. Impaired sensory handling is apparent in our individuals, supported by the incidence of impaired gait, small, cold hands/feet, abnormal tone, and hypoalgesia, and all individuals who reported self-harm behaviors had one of these sensory processing issues (tone abnormalities, temperature mishandling, diminished response to pain), so there may be a correlation between self-harm and impaired sensory handling. 

Another notable result was the presence/absence of developmental delay and regression in our individuals. All but one of our individuals had developmental delays. This subject had the least number of major or minor criteria met in the whole study, meeting only 1 supportive criterion (abnormal muscle tone), and no major criteria. This individual is a twin, who carries a pathogenic variant outside the nuclear localization sequence which is part of the *HNRNPH2* gene where most individuals carry pathogenic variants, however Kreienkamp et al. expanded the clinical spectrum including males carrying pathogenic variants in *HNRNPH2* also showing that variants outside this region may not present with as severe phenotypes [22]. Seventeen individuals reported regression followed by stabilization or recovery. As stated above, regression followed by stabilization or recovery is a necessary criterion for both Typical Rett Syndrome diagnosis and Atypical Rett Syndrome diagnosis, however, none of the 3 individuals previously diagnosed with Rett Syndrome reported regression in speech, motor, or cognitive skills. 

Finally, we noted that 14 individuals reported comorbid seizures. Seizures are one of the most common phenotypes accompanying neurodevelopmental disorders and was previously reported in 39% of individuals with *HNRNPH2-*neurodevelopmental disorder [16]. Notably, seizures are quite common in individuals with RTT, as the literature reports that 59% of a cohort of 389 RTT patients had seizures, with 48% of the 389 reporting having a diagnosis of epilepsy [24]. More studies are warranted to elucidate the molecular pathways underlying seizure pathophysiology in *HNRNPH2*-related NDD and RTT [16,17,19,21,22,23,24,25,26].

We suggest *HNRNPH2-*RNDD can lead to an RTT-like clinical phenotype. There are several limitations to this study. First, we have a small total number of survey responses, with only 26 being included for the final analysis. In rare diseases, it is difficult to capture a large “n”, however this survey was completed by about 50% of the total number of individuals in the registry at that time. In addition, this is a parent or caregiver-reported survey which has inherent selection bias in addition to potential recall bias. We did not correlate the surveys with any standardized clinical evaluation or assessment. Additionally, upon review of study results, we realize that multiple choice answers limited the caregivers’ ability to provide more qualitative feedback in response to the questions. Future surveys will assure to include a variety of question types to address this concern. Lastly, part of the exclusion criteria proposed by Neul et al. requires an individual to lack “gross psychomotor delay” in the first six months of age, which is not well described in the diagnostic clinical criteria. While 25/26 individuals in our cohort presented with developmental delays, we were not able to capture what the severity of their motor developmental delays were by 6 months of age. The prospective arm of the NHS will hopefully provide clarification by including additional standardized measures.

## 5. Conclusions

In summary, this study highlights some of the common clinical features of *HNRNPH2-*related-NDD which are also seen in RTT Syndrome. We suggest clinicians consider *HNRNPH2*-related-NDD on their differential diagnosis when evaluating patients presenting with Rett-like phenotypes and perhaps including *HNRNPH2* on genetic testing panels for Rett presentations. By including *HNRNPH2* in these clinical decision-making algorithms, more individuals will likely be diagnosed with *HNRNPH2*-related NDD and possibly at earlier ages. Lastly, we are eager to further characterize the motor stereotypies in this cohort and better understand the underlying pathophysiology of these motor abnormalities.

## Figures and Tables

**Figure 1 genes-14-01154-f001:**
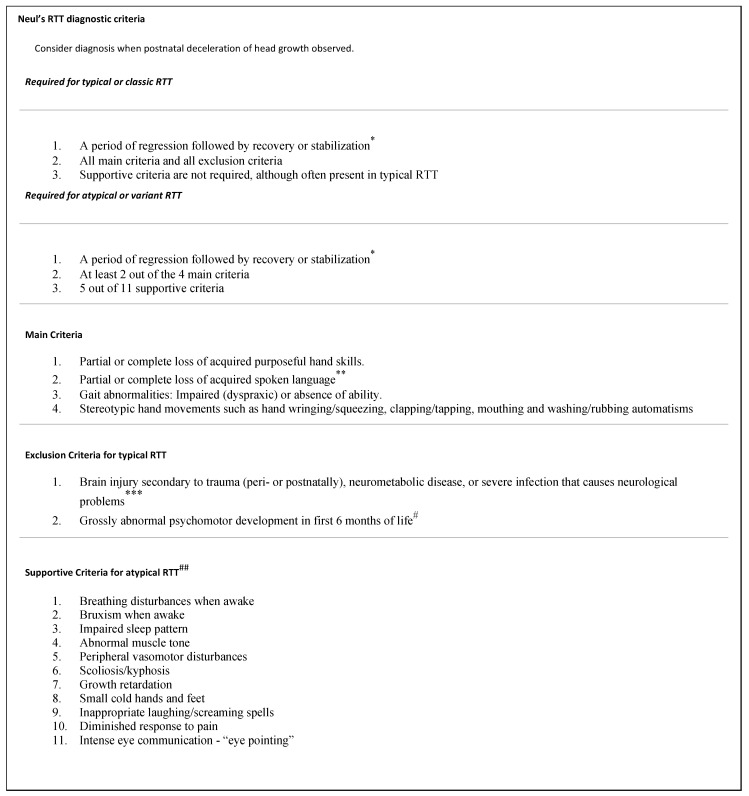
Neul’s criteria. Adapted directly from Neul’s 2011 work, listed in Works Cited. Adapted with permission from *Annals of Neurology-Wiley Online Library.* * Because *MECP2* mutations are now identified in some individuals prior to any clear evidence of regression, the diagnosis of “possible” RTT should be given to those individuals under 3 years old who have not lost any skills but otherwise have clinical features suggestive of RTT. These individuals should be reassessed every 6–12 months for evidence of regression. If regression manifests, the diagnosis should then be changed to definite RTT. However, if the child does not show any evidence of regression by 5 years, the diagnosis of RTT should be questioned. ** Loss of acquired language is based on best acquired spoken language skill, not strictly on the acquisition of distinct words or higher language skills. Thus, an individual who had learned to babble but then loses this ability is considered to have a loss of acquired language. *** There should be clear evidence (neurological or ophthalmological examination and MRI/CT) that the presumed insult directly resulted in neurological dysfunction. # Grossly abnormal to the point that normal milestones (acquiring head control, swallowing, developing social smile) are not met. Mild generalized hypotonia or other previously reported subtle developmental alterations during the first six months of life is common in RTT and do not constitute an exclusionary criterion. ## If an individual has or ever had a clinical feature listed it is counted as a supportive criterion. Many of these features have an age dependency, manifesting and becoming more predominant at certain ages. Therefore, the diagnosis of atypical RTT may be easier for older individuals than for younger. In the case of a younger individual (under 5 years old) who has a period of regression and ≥2 main criteria but does not fulfill the requirement of 5/11 supportive criteria, the diagnosis of “probably atypical RTT” may be given. Individuals who fall into this category should be reassessed as they age, and the diagnosis revised accordingly.

**Table 1 genes-14-01154-t001:** Demographic Table Part 1 (Note: For average age, the two deceased individuals were included in our analysis).

Average age of Individuals (years)	14.4
Sex of patients	23 females, 3 males
Total genotypes represented	7
Most common genotypes	p.Arg206Trp
Ethnicities represented	3
Ethnicities reported	Caucasian (n = 23), Hispanic (n = 2), East Asian (n = 1)

**Table 2 genes-14-01154-t002:** Proportions of each phenotype. Each proportion is out of the total number of individuals who provided information about that specific phenotype. The total may not add up to 26 if not all individuals provided information about the phenotype listed.

Phenotypes (Neul’s Major Criteria Bolded)	Proportions of Individuals with Phenotypes
**Partial or complete purposeful loss of hand skills**	1/26 (4%)
**Partial or complete loss of acquired spoken language**	5/26 (19%)
**Gait abnormalities**	21/26 (81%)
**Hand stereotypies**	18/26 (69%)
Breathing abnormalities	5/26 (19%)
Bruxism while awake	12/26 (46%)
Impaired sleeping patterns	6 /25 (24%)
Impaired muscle tone (hypo/hypertonia)	25/26 (96%)
Small, cold hands/feet	13/26 (50%)
Vasomotor disturbances	8 /26 (31%)
Scoliosis or kyphosis	10/26 (38%)
Growth Retardation	9 /25 (36%)
Inappropriate laughing/screaming spells	15/26 (58%)
Diminished response to pain	13/24 (54%)
“Eye pointing”	6/26 (23%)

**Table 3 genes-14-01154-t003:** Proportions of individuals who met the number of Neul’s criteria described above. For example, 4 out of our 26 patients met 3 of Neul’s main criteria, and 5 or more supportive criteria.

4 main criteria	0
3 main criteria, 5 or more supportive criteria	4/26 (15%)
3 main criteria, less than 5 supportive criteria	1/26 (4%)
2 main criteria, 5 or more supportive criteria	8/26 (31%)
2 main criteria, less than 5 supportive criteria	5/26 (19%)
1 main criteria, 5 or more supportive criteria	2/26 (8%)
1 main criteria, less than 5 supportive criteria	2/26 (8%)
No main criteria, 5 or more supportive criteria	0
No main criteria, less than 5 supportive criteria	4/26 (15%)

## Data Availability

All data available upon request to the authors. Please email jb3634@cumc.columbia.edu for any and all requests.

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
