# Peer review of "Rett-like Phenotypes in HNRNPH2-Related Neurodevelopmental Disorder"

_genes, 2023, doi:10.3390/genes14061154_

Round 1

Reviewer 1 Report

The MS by Gonzalez et al covers an interesting topic on a Rett (RTT)-like phenotype in n=33 pts harboring de novo pathogenic missense variants X-linked HNRNPH2 gene mutations. Phenotypic features include developmental delay, ID, seizures, autistic-like features, and motor abnormalities. The AA further characterized RTT clinical features in this group of individuals by use of caregiver reports. The phenotypic overlap with atypical RTT syndrome is evidenced and should prompt clinicians to consider HNRNPH2 on the differential diagnostics of atypical RTT.

Comments

1. Table on Major and Minor RTT criteria could be redundant as they are well known in the field.

2. 1946 is unlikely as the first date for Andreas Rett's publication (1966).

Author Response

  1. Table on Major and Minor RTT criteria could be redundant as they are well known in the field.

We thought the addition of this table was essential for those not familiar with Neul’s criteria to make the paper stronger, as Rett Syndrome is a clinically described entity with specific clinical criteria.  Having this table allows for easier readability.

  1. 1946 is unlikely as the first date for Andreas Rett's publication (1966).

We have edited this as recommended.  We thank the reviewer for bringing our attention to this very unfortunate typo, that we have now corrected.

Reviewer 2 Report

1. In the Abstract, in "Methyl-CpG Binding Protein 2 (MECP2)", the gene symbol MECP2 should be presented in italics. All other gene citations in the text, such as CDKL5, FOXG1 and HNRNPH2, must all be presented in italics.  

2. I suggest authors to include in the Discussion section of the manuscript a brief discussion of HNRNPH2-associated atypical Rett-like syndrome and the comparison of clinical, genetic and neuroimaging aspects with other genetic aspects and subtypes related to atypical Rett syndrome and Rett-like syndrome. Some references which can aid in this step include: Mol Genet Genomic Med 2019;7(11):e968 ; Curr Opin Psychiatry 2018;31(2):103-108. 

3. There are also some minor typos which should be corrected: 

- Page 1, Line 42: "MEPC2" should be changed to MECP2.

- In several paragraphs, "et. al" should be changed to "et al.". 

Author Response

Reviewer #1

  1. In the Abstract, in "Methyl-CpG Binding Protein 2 (MECP2)", the gene symbol MECP2 should be presented in italics. All other gene citations in the text, such as CDKL5, FOXG1 and HNRNPH2, must all be presented in italics.  

We have edited this as recommended.  Every gene name has been updated to an italicized font.

  1. I suggest authors to include in the Discussion section of the manuscript a brief discussion of HNRNPH2-associated atypical Rett-like syndrome and the comparison of clinical, genetic and neuroimaging aspects with other genetic aspects and subtypes related to atypical Rett syndrome and Rett-like syndrome. Some references which can aid in this step include: Mol Genet Genomic Med 2019;7(11):e968 ; Curr Opin Psychiatry 2018;31(2):103-108. 

We have edited this as recommended for FOXG1 & CDKL5, and we compared the results we found to the findings in HNRNPH2-related neurodevelopmental disorder patients. 

  1. There are also some minor typos which should be corrected: 

- Page 1, Line 42: "MEPC2" should be changed to MECP2.

- In several paragraphs, "et. al" should be changed to "et al.". 

We have edited this as recommended.  Spelling changes have been made.

Reviewer 3 Report

In this study, the authors sought to determine presence of clinical features of Rett syndrome in individuals with HNRNPH2-related neurodevelopmental disorder by using caregiver report. The study involved 33 individuals with HNRNPH2-related neurodevelopmental disorder, 26 of which met inclusion criteria for final analysis. The study found that caregivers reported a high number of signs/symptoms that are part of Rett syndrome clinical diagnostic criteria, including some of the major criteria of the syndrome. Based on the survey results, 12 individuals met the diagnostic criteria for atypical Rett syndrome. The findings suggest that HNRNPH2-related neurodevelopmental disorder exhibits clinical characteristics that overlap with those of RTT and that HNRNPH2 should be considered in the differential diagnosis of atypical Rett syndrome.

Overall, the article is well written and clear. Provides good background information regarding clinical diagnosis of Rett syndrome and the differentiating characteristics of typical vs atypical Rett syndrome.

There are some methodological shortcomings that prevent this from being a meaningful study. The use of caregiver completed survey as a means of diagnosis is problematic as is acknowledged in the limitations discussion. Importantly, many of the clinical symptoms listed in the diagnostic criteria are highly technical in nature and depending on wording of the questions, caregivers unfamiliar with medical terminology would not be able to accurately assess presence/absence without guidance by an experienced clinician. The authors do not provide a description of how the survey questions were phrased or comment whether there was any direct contact to help caregivers complete the survey. Additionally, the clinical diagnostic criteria for Rett syndrome were derived based on in-person assessments performed by clinicians highly experienced with managing Rett syndrome thus the generalizability/validity in a remote survey format is questionable.

The clinical utility of the findings is also questionable. Neurologists and geneticists are well aware of the limitations of clinical phenotyping. It is well-described that a diagnosis of atypical Rett syndrome is non-specific as to genetic etiology therefore would require further testing with exome or genome sequencing to cast a broad net on all possible genetic etiologies. The suggestion to focus-in specifically on HNRNPH2-related neurodevelopmental disorder seems counterintuitive and unhelpful from a diagnostic perspective.

The findings would certainly have greater validity if the cohort were to be assessed in-person by clinicians familiar with Rett syndrome diagnostic criteria, but even then the significance and clinical utility of the findings would remain in question.

Author Response

Reviewer #2:

There are some methodological shortcomings that prevent this from being a meaningful study. The use of caregiver completed survey as a means of diagnosis is problematic as is acknowledged in the limitations discussion. Importantly, many of the clinical symptoms listed in the diagnostic criteria are highly technical in nature and depending on wording of the questions, caregivers unfamiliar with medical terminology would not be able to accurately assess presence/absence without guidance by an experienced clinician. The authors do not provide a description of how the survey questions were phrased or comment whether there was any direct contact to help caregivers complete the survey. Additionally, the clinical diagnostic criteria for Rett syndrome were derived based on in-person assessments performed by clinicians highly experienced with managing Rett syndrome thus the generalizability/validity in a remote survey format is questionable.

We understand the reviewers concern about these limitations of our study methodology. We appreciate that a parent-reported survey is certainly not applied as diagnostic clinical criteria and did not intend to do so in this study. We have clarified in the discussion that our intention was to highlight some of the caregiver-reported signs and symptoms that may overlap with a clinician-utilized survey for diagnosis. In addition, we have clarified that study team who created the questionnaire were clinicians who have experience with Rett patients, including a pediatric neurologist, clinical geneticist and developmental psychologist. However, these individuals assisted in study design, and did not intend to utilize the survey for in person evaluations and diagnostic purposes. Again, we certainly appreciate the significance of this comment and hope the reviewers find our revisions appropriate to dismiss any readers’ concern about the use of this sort of survey in our study and in further diagnostic situations.

The clinical utility of the findings is also questionable. Neurologists and geneticists are well aware of the limitations of clinical phenotyping. It is well-described that a diagnosis of atypical Rett syndrome is non-specific as to genetic etiology therefore would require further testing with exome or genome sequencing to cast a broad net on all possible genetic etiologies. The suggestion to focus-in specifically on HNRNPH2-related neurodevelopmental disorder seems counterintuitive and unhelpful from a diagnostic perspective. The authors are most appreciative and supportive of this comment! The authors propose that this study is merely another example of identifying a clinical phenotype with a newly described genetic etiology.  We are certainly not suggesting that readers should specifically think of HNRNPH2 as the sole (or one of the few) genotype when thinking of atypical RTT, we are suggesting that HNRNPH2 should be included on any gene panels or clinical phenotype descriptions for genetic testing which is more commonly being used in an array of neurodevelopmental disorders. The authors have modified the text to assure this comment is described fully in the text revision.

The findings would certainly have greater validity if the cohort were to be assessed in-person by clinicians familiar with Rett syndrome diagnostic criteria, but even then the significance and clinical utility of the findings would remain in question. Again, the authors are in agreement with this comment! The authors are continuously meeting with families affected by this rare disorder, and this initial case series was to bring about awareness of the clinical phenotype. The authors are planning to include standardized, validated prospective testing for these features with  in-person assessments at future clinic evaluations.

Round 2

Reviewer 3 Report

There additional clarifying statements in the text are beneficial.